# Virtual Reality Gaming Elevates Heart Rate but Not Energy Expenditure Compared to Conventional Exercise in Adult Males

**DOI:** 10.3390/ijerph16224406

**Published:** 2019-11-11

**Authors:** Théo Perrin, Charles Faure, Kévin Nay, Giammaria Cattozzo, Anthony Sorel, Richard Kulpa, Hugo A. Kerhervé

**Affiliations:** 1Univ Rennes, Inria, M2S - EA 7470, F-35000 Rennes, France; theo.perrin@ens-rennes.fr (T.P.); anthony.sorel@univ-rennes2.fr (A.S.); richard.kulpa@univ-rennes2.fr (R.K.); 2Univ Rennes, M2S - EA 7470, F-35000 Rennes, France; kevin.nay@inra.fr (K.N.); giammariacattozzo@gmail.com (G.C.)

**Keywords:** active gaming, fitness, virtual reality, energy expenditure, heart rate

## Abstract

Virtual reality using head-mounted displays (HMD) could provide enhanced physical load during active gaming (AG) compared to traditional displays. We aimed to compare the physical load elicited by conventional exercise and AG with an HMD. We measured energy expenditure (EE) and heart rate (HR) in nine healthy men (age: 27 ± 5 years) performing three testing components in a randomised order: walking at 6 km/h (W6), AG, and AG with an additional constraint (AG^W^; wrist-worn weights). Although we found that HR was not significantly different between W6 and the two modes of AG, actual energy expenditure was consistently lower in AG and AG^W^ compared to W6. We observed that playing AG with wrist-worn weights could be used as a means of increasing energy expenditure only at maximum game level, but ineffective otherwise. Our findings indicate that AG in an HMD may not provide a sufficient stimulus to meet recommended physical activity levels despite increased psychophysiological load. The differential outcomes of measures of HR and EE indicates that HR should not be used as an indicator of EE in AG. Yet, adding a simple constraint (wrist-worn weights) proved to be a simple and effective measure to increase EE during AG.

## 1. Introduction

Insufficient physical activity is one of the main risk factors of non-communicable diseases (cardiovascular, neurodegenerative or metabolic diseases) [1]. More than 30% of people aged 15 years and older do not currently meet recommended physical activity levels (150 min of moderate-intensity activity per week for adults 18–64 yr) [1]. In this context, increasing physical activity not only through exercise, but also for commuting, during work and leisure, are often recommended as effective measures to curb the development of chronic illnesses linked with physical inactivity.

While media usage such as video games do not provide adequate physical strain [2,3], combining video games and physical activity or motor skills (active gaming or exergaming) has been used for close to 20 years to contribute meeting physical activity recommendations [4]. Active gaming (AG) has been reported to improve mood and enjoyment compared to conventional forms of exercise like walking [5,6,7]. As such, AG can also be used to increase motivation and contribute to increase adherence to training regimen [8], for instance by creating competitive situations between different participants [9] or providing feedback from previous performances [8]. Clinical populations, especially individuals with functional limitations have shown improvements in energy efficiency and expenditure using AG [10]. However, in healthy populations, most forms of AG have not been found to elicit levels of energy expenditure comparable to conventional exercises (walking, cycling) [5,6,11,12] and therefore, may not be suitable for health promotion [13,14]. Only whole-body AG performed on various commercially-available platforms has been shown to elicit substantial energy expenditure ranging −18–30 kJ/min·kg, matching or even exceeding levels of energy expenditure induced by treadmill walking at −5 km/h [15].

One common feature of all aforementioned studies is that they have used TV or computer screens. Recent technological developments displaying virtual reality (VR) environments using head-mounted displays (HMD) with stereoscopic vision and large field of view have the potential to further increase feelings of immersion and sensory-motor experience compared with 2-D screens [16]. For instance, performance and enjoyment during a high-intensity interval workout improved when performed within a virtual environment using an HMD in young and healthy participants, and further increased when adding an avatar based on a previously achieved performance within the same virtual environment [17].

However, despite the rapid development of commercially-available AG in HMD, no studies have evaluated whether the greater immersive feelings provided by HMD during AG elicit sufficient levels of physical activity. Therefore, the main purpose of this study was to quantify and compare the physiological load elicited by conventional walking exercise with a whole-body, to a commercially-available AG in an HMD. We sought to compare measures of EE, as a measure of true physiological load, to heart-rate (HR), as a general indicator of the total psycho-physiological load experienced by players, to determine whether AG in HMD truly increased energy expenditure or merely increased stress from a heightened emotional engagement in the game. To further highlight any potential discrepancy between EE and HR, we included an additional condition known to promote EE simply by adding weight at the wrists [18]. Finally, because most AG are designed using levels of incremental difficulty, a secondary aim of this study was to assess at which level, if any, sufficient levels of physical activity were achieved, and thus provide a basis for future AG developments in HMD.

## 2. Materials and Methods

Nine healthy adult males (age: 27 ± 5 yr, height: 177 ± 5 cm, weight: 76 ± 13 kg) with a weekly exercise training time ranging <2.5 to 15 h/wk volunteered for this study. In accordance with the Helsinki Declaration of 1975, as revised in 2008, participants received an information sheet describing study procedures, risks associated with the testing, and were informed they could withdraw at any moment without comment or penalty. All participants gave their written consent prior to inclusion in the study, and their results were anonymised before analysis.

### 2.1. Study Procedures

All procedures were performed in a controlled laboratory environment (−21 °C, 60% relative humidity). An initial habituation phase was implemented to control for the effects of gaming expertise. Participants were free to practice until they felt comfortable with the game and HMD (10–30 times) with the game used during testing. Following this habituation phase, participants reported to the laboratory for a single session lasting −2.5 h euhydrated and having abstained from alcohol and caffeine for 12 h.

During this testing session, participants performed three tests in a randomized order: a submaximal walking test on a treadmill at 6 km/h for 15 min (W6), one bout of active gaming with HMD at body weight (AG), and one bout of active gaming with HMD and wrist-worn weights (−2.5% of body weight; AG^W^). Prior to the first condition, a 5 min resting period (seating) was observed for baseline measurements, and all conditions were interspersed with resting periods ensuring full recovery (heart rate and oxygen consumption within 15% of resting values).

The virtual reality game used in this study was a free-to-play bow shooting game (Longbow, The Lab game suite, Valve, Bellevue, WA, USA) for Vive VR headset OPJT100 (HTC Corporation; Taiwan). The purpose of the game is to defend the entrance of a castle by eliminating successive waves of opponents with increasing difficulty level (successive waves with increasing number of opponents and/or frequency), which determined the level of the game achieved. Difficulty and its evolution were internal parameters of the game, and no modification was made by experimenters. Performance in AG and AG^W^ was determined by the level reached and the number of actions performed at each level (successful and unsuccessful), which were continuously monitored by one of the investigators using a purpose-build software (Unity, San Francisco, CA, USA). The game ended when opponents entered through the main door of the castle.

### 2.2. Measurements

Heart rate (HR) was monitored continuously during W6, AG and AG^W^ and recovery periods, using a portable 12-lead electrocardiogram (X12+, Mortara, Milwaukee, WI, USA). Individual age-predicted maximal HR (HR_max_) was determined using Equation (1) [19] and HR was expressed as a percentage of HR_max_ using Equation (2):HR_max_ = 208 − 0.7 × (age in year)(1)

HR% = (HR × 100)/HR_max_(2)

Expired gases at the mouth were continuously measured during W6, AG, and AG^W^ using an automated, breath-by-breath indirect calorimetry system (Ultima CPX, Medgraphics, St. Paul, MN, USA). The metabolic cart was located outside of the reach of participants using a 1.5 m sampling hose, ensuring full range of movement and adding only trivial weight (neoprene mask, sampling tube and hose). Relative energy expenditure (EE, in kJ/kg·h) was subsequently calculated in the standard manner using minute relative oxygen consumption (VO_2_) and respiratory exchange ratio (RER), based on the energy equivalent of oxygen (1 L of oxygen yielding 19.6–21.1 kJ for RER ranging 0.7–1.0), as described in Equation (3) [20]:EE = VO_2_ × (5 × RER + 16.1) × 60(3)

### 2.3. Statistical Analyses

Data are expressed as mean ± standard deviation (SD), and were initially tested for normal distribution using the Shapiro–Wilk W-test. All statistical analyses were performed using the open-access statistical package jamovi [21], and the level of significance set at *p* < 0.05. A repeated-measures (RM) design was chosen for this study to capture the range of physical abilities in the participant sample. One-way, repeated measures ANOVAs with Bonferroni post hoc were used to assess the effect of mode (rest, W6, AG, AG^W^) on HR% and EE. Two-way, repeated measures ANOVAs with Bonferroni post hoc tests were used to assess the effect of game level, added weight, and their interaction, on HR% and EE. Pairwise comparisons between AG and AG^W^ (game duration, number of actions performed, time spent playing at HR% and EE above W6 values) and between W6 and AG and W6 and AG^W^ were tested for significant differences using Student’s *t*-test (using HR% and EE at maximum level attained).

Effect sizes for pairwise comparisons were assessed using Cohen’s d, calculated in the standard manner and interpreted according to Cohen’s scale (small effect: 0.2 < *d* < 0.5, moderate effect: 0.5 < *d* < 0.8, and large effect: *d* > 0.8). For ANOVAs, we reported effect sizes using partial eta-squared (η^2^p) interpreted according to Cohen’s scale (small effect: 0.01 < η^2^p < 0.06, medium effect: 0.06 < η^2^p < 0.14, and large effect: η^2^p > 0.14).

## 3. Results

### 3.1. General Results

All participants completed the three components of the testing session. There was a significant main effect of mode with large practical effect for average HR% (Table 1), with post hoc testing revealing all exercise conditions (AG, AG^W^, and W6) induced significantly greater HR% values than rest (*p* < 0.001), a higher average HR% in AG^W^ compared to AG (*p* = 0.004), and no significant differences between W6 and AG (*p* = 0.187), and between W6 and AG^W^ (*p* = 0.739). There was also a significant main effect of mode with large effect size for average EE (Table 1), with post hoc testing revealing all exercise conditions (AG, AG^W^ and W6) induced significantly greater EE values than rest (*p* < 0.001), and W6 inducing a greater average EE than both AG and AG^W^ conditions (*p* < 0.001), with no significant difference between AG and AG^W^ (*p* = 0.079).

### 3.2. Effect of Game Level

All participants reached at least the fourth level of the game, and there were no significant differences between the two game conditions in duration (AG: 14.6 ± 0.7 min; AG^W^: 14.7 ± 0.9 min; *p* = 0.807; *d* = 0.08) or number of actions performed (AG: 423 ± 15; AG^W^: 420 ± 71; *p* = 0.91; *d* = −0.37). There were significant and large main effects of game level (*p* < 0.001, η^2^p = 0.914), added weights (*p* < 0.001, η^2^p = 0.782) and interaction (*p* = 0.008, η^2^p = 0.342) on HR% (Figure 1). Post hoc revealed differences in HR% between AG and AG^W^ at level 4 and maximum level reached (*p* < 0.001) (Figure 1). There were also significant and large main effects of game level (*p* < 0.001, η^2^p = 0.951), added weights (*p* < 0.001, η^2^p = 0.641) and interaction (*p* < 0.001, η^2^p = 0.517) on EE (Figure 2). Post hoc tests revealed differences between AG and AG^W^ only at maximum level reached (*p* < 0.001) (Figure 2).

There were significant differences in HR% at W6 and at maximum AG level (*p* < 0.001, *d* = 1.749), as well as at maximum AG^W^ level (*p* < 0.001, *d* = −3.221), with a total time spent above W6 longer in AG^W^ than AG (7.08 ± 3.9 min vs. 3.06 ± 3.9 min; *p* = 0.004; *d* = 2.146). There were no significant differences in EE at W6 and at maximum level during AG (*p* = 0.146, *d* = 0.537) or AG^W^ (*p* = 0.083, *d* = 0.661), and no significant differences in total time spent above W6 between AG^W^ and AG (0.61 ± 0.8 min vs. 0.11 ± 0.2 min; *p* = 0.07; *d* = 0.65).

## 4. Discussion

The current study permitted to quantify and compare the physiological load (as measured by EE and HR%) of conventional exercise and AG using HMD, and determine its usefulness as a tool to contribute fighting against the effects of physical inactivity on health.

In this study, we observed that AG induced increases in relative heart rate similar to that of a brisk walk compared to rest. However, the main finding of this study was that average EE during the −15 min bout of AG was consistently inferior to walking EE. We initially sought to compare responses in EE and HR to determine whether AG in HMD truly increased physiological load, since, unlike EE, HR can be greatly influenced by both psychological and physiological factors. Therefore, from our findings, it may be inferred that AG in HMD resulted in a greater psycho-physiological strain without accompanying increases in relative energy expenditure. Similar results have been observed in sedentary video games played on 2-D displays with a strong cognitive load, for instance in a racing game raising HR without movement [2]. Therefore, the differential outcomes of measures of HR and EE in the current study indicates that HR, as a global measure of psycho-physiological strain, should not be used as an indicator of EE in AG. It is also paramount the psychological and emotional aspects of AG in HMD must be evaluated to fully apprehend the balance of health benefits and risks for the prescription of active gaming.

More specifically, we found that the AG in HMD could elicit levels of physical activity typically found during conventional exercise such as walking, only very briefly, during AG at the maximum level attained. Overall, it is clear that the kind of AG used in this study did not elicit sufficient levels of physical activity, such as may be used to improve health outcomes compared to conventional exercise [14], and might play a role in the fight against physical inactivity [4].

Surprisingly, we also found that playing AG with added weights provoked an increase in EE only at advanced game levels, but not on average: the additional physiological load induced by wearing weight on wrists in the AG^W^ condition (+17% in EE and +14% in HR% on average) increased as a function of game level (from 16% to 22% increase in EE and from 11% to 16% increase in HR% from levels 1–4). However, adding weights at the wrists induced EE comparable to W6 only at higher AG levels, and further underscored the comparatively greater response in HR% than EE. Still, we observed that adding wrist-worn weights induced a large change in the time spent at or above walking EE, especially at the higher levels of the game, and thus, this simple physical constraint could be used increase the intensity of physical exercise during AG as is commonly done for conventional exercise [18]. Therefore, future studies could evaluate the specific role of task-related constraints (rules changes, feedback, cooperative vs. competitive situations) or modified environmental conditions (ambient temperature, oxygen partial pressure) to increase energy expenditure for health promotion purposes.

Finally, although the AG we used was a whole-body exercise, it did not involve rhythmic activity from large muscle masses and required only quick shuffling to change positions. Following previous endeavours in AG with 2-D [15] or 3-D displays [22], future AG designs in HMD should include locomotor tasks or even defensive actions to promote increased energy expenditure. Our findings are also of interest for guiding the development of AG specifically for the health sector as alternative forms of rehabilitation for clinical populations with measurable physical [23] and psychological benefits [7].

## 5. Conclusions

The findings of the current study indicate that the enhanced immersive feelings provided during active gaming in a head-mounted display were not sufficient to meet recommended physical activity levels. The differential outcomes of measures of relative heart rate and relative energy expenditure indicates that heart rate, as a global measure of psycho-physiological strain, should not be used as an indicator of energy expenditure in active gaming designed for increasing levels of physical activity. Surprisingly, adding a simple constraint (wrist-worn weights) during active gaming was effective to increase energy expenditure at levels comparable to conventional exercise only at the highest level of the game.

## Figures and Tables

**Figure 1 ijerph-16-04406-f001:**
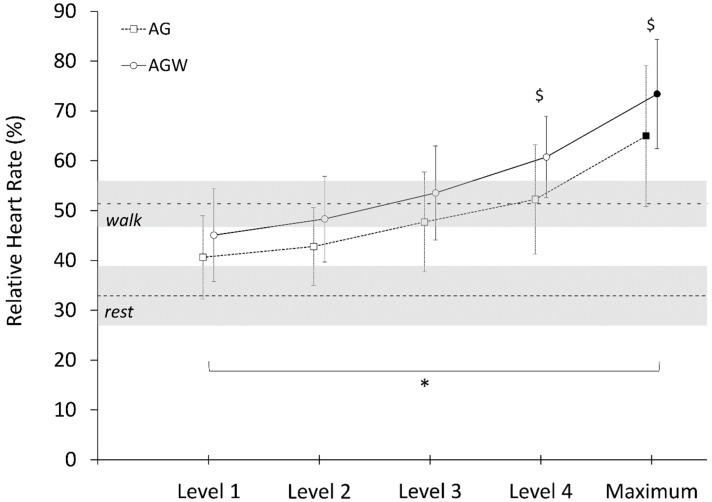
Relative heart rate (%, expressed relative to individual-predicted maximum) across game levels (1–4) and maximum level attained (Maximum) in active gaming with (AG) and without added weights (AG^W^). Rest and walk (W6) average values are represented in dashed lines with ± 1 standard deviation (greyed area). Symbol * represents the main significant effect of exercise (*p* < 0.001), and symbol $ represents significant differences between AG and AG^W^ (*p* < 0.001).

**Figure 2 ijerph-16-04406-f002:**
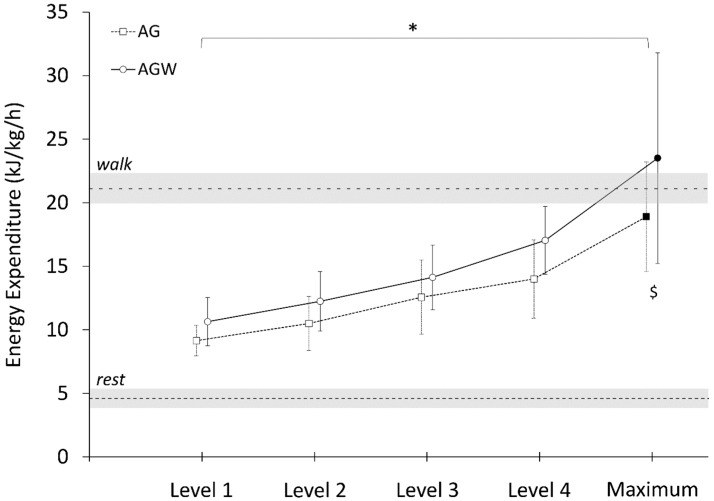
Relative energy expenditure (EE) across game levels (1–4) and maximum level attained (Maximum) in active gaming with (AG) and without added weights (AG^W^). Rest and walk (W6) average values are represented in dashed lines with ± 1 standard deviation (greyed area). Symbol * represents the main significant effect of exercise, and symbol $ represents significant differences between AG and AG^W^ (*p* < 0.05).

**Table 1 ijerph-16-04406-t001:** Group average for relative heart rate (%HR) and relative energy expenditure (EE) during rest, active gaming (AG), active gaming with added weights (AG^W^), and walk (W6).

	Rest	W6	AG	AG^W^	*p*-Value	η^2^p
HR%	33 ± 6	51 ± 5 ^A^	47 ± 5 ^A^	54 ± 4 ^A C^	<0.001	0.883
EE (kJ/kg·h)	4.6 ± 0.7	21.1 ± 1.2 ^A^	11.7 ± 2.5 ^A B^	13.7 ± 4.9 ^A B^	<0.001	0.954

^A^ Significantly different to Rest (*p* < 0.001) ^B^ Significantly different to W6 (*p* < 0.001) ^C^ Significantly different to AG (*p* < 0.05).

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
