# Peer review of "Virtual Reality Gaming Elevates Heart Rate but Not Energy Expenditure Compared to Conventional Exercise in Adult Males"

_ijerph, 2019, doi:10.3390/ijerph16224406_

Round 1
Reviewer 1 Report
This study examined the effects of active gaming (AG) compared to walking and active gaming with hand weights. The study examined heart rate and energy expenditure (EE). The premise or hope was that AG could increase HR and EE similar to walking for the beneficial health effects.
All abbreviations must be explicitly stated in abstract and manuscript. This reviewer does not believe AG was stated.
There are a lot of abbreviations in the manuscript. A table or chart with all of them may be positive.
Adult males should be in title.
Heart rate (HR) may or may not be associated with all feelings. This is bit of a stretch that needs reframed throughout the manuscript. Same as psycho-physiological strain. This wasn’t directly measured.
Why was RER not reported?
What is the energy expenditure equation? Seems like wrong citation?
Please present HR as well as HR%. What is Relative Heart Rate? Is this same as table? It is described in figure but not methods.
Length of time isn’t mentioned. Average response would depend on length of time and level? This wasn’t covaried although there was no difference on average. How long was the walk?
Please report all groups in figures.
Inclusion of cohen’s d is positive.
Author Response
RESPONSES TO REVIEWERS
Please find enclosed our new manuscript taking into account all the recommendations and suggestions of the reviewers. We intended to answer the most precisely as possible to reviewers’ questions. The authors would like to thank the reviewers for their recommendations and suggestions which has helped to greatly enhance the quality of this manuscript.
Reviewer #1 (Comments to the Author (Required)):
All abbreviations must be explicitly stated in abstract and manuscript. This reviewer does not believe AG was stated. There are a lot of abbreviations in the manuscript. A table or chart with all of them may be positive.
We agree with the reviewer. We have clarified the abbreviations in the document and have added a table with all abbreviations to make the document clearer.
Adult males should be in title.
Thanks for your suggestion. After discussion within the group, we agree with the reviewer and have modified the title accordingly.
Heart rate (HR) may or may not be associated with all feelings. This is bit of a stretch that needs reframed throughout the manuscript. Same as psycho-physiological strain. This wasn’t directly measured.
We thanks the reviewer for their astute remark. It is true that we did not measure cognitive strain or stress. However, heart rate is under the influence of both psychological and physiological stressors, and therefore, we feel strongly that it is a good indicator of the total psycho-physiological load experienced by participants. Accordingly, we have carefully reworded sections dealing with this variable to increase accuracy. We hope this justification suits the reviewer.
Why was RER not reported?
The measure of RER in and of itself was not a variable of interest for our study. We used RER in the calculation of EE. As such, we chose not to report it as a result for clarity's sake.
What is the energy expenditure equation? Seems like wrong citation?
The energy expenditure equation we used is a standard calculation in exercise physiology using the linear regression of EE as a function of RER and VO2 (with a y=ax+b form with known energy equivalents of lipids and glucids oxidized at different RER levels). We have added the reference of the princeps study on substrate utilization as a function of RER. We have updated the manuscript accordingly.
Please present HR as well as HR%. What is Relative Heart Rate? Is this same as table? It is described in figure but not methods.
Thanks for your suggestion, we have now included results of HR as well. HR% is heart rate expressed to the age-predicted participants' maximum, according to the Tanaka reference. It allows more accurate comparisons across participants.
Length of time isn’t mentioned. Average response would depend on length of time and level? This wasn’t covaried although there was no difference on average. How long was the walk?
Thanks for your remark, it is true this section could have been made clearer. We have added information about the time spent for each trials in the methods.
Please report all groups in figures.
We have only 1 group in this study. Accordingly, our figures describe the results of the whole group for EE and HR under different conditions (Rest; Walk 6km.h-1; Active gaming and Active gaming with wrist-worn weights).
Inclusion of cohen’s d is positive.
We would like thank the reviewer for this positive comment, and again, for the time and energy spent on our manuscript.
Reviewer 2 Report
Hello,
I appreciate the opportunity reviewing the work, I think all of you enjoyed doing it, and I hope that this worked had fostered a will for the science.
The attached file contains highlighted sentences throughout the text, as well ass comments within, in which I tried to give my opinion and suggestions, questioning as well, in order to improve you work.
Best regards

Author Response
RESPONSES TO REVIEWERS
Please find enclosed our new manuscript taking into account all the recommendations and suggestions of the reviewers. We intended to answer the most precisely as possible to reviewers’ questions. The authors would like to thank the reviewers for their recommendations and suggestions which has helped to greatly enhance the quality of this manuscript.
Reviewer #2 (Comments to the Author (Required)):
Line 13 : "AG with" - active gaming (AG)
Thanks for your comment, we have amended the manuscript.
Line 14 : "in nine healthy" - Age would be helpful here.
Thanks for your comment, we have amended the manuscript.
Line 56 : "physical activity" - There are some studies related to Nintendo Wii and rehabilitation in stroke, mental palsy and others that could be at least cited here or in discussion (didn't read it yet) that may serve as a base for game-based exercises/rehabilitation
Thanks for the suggestion, we agree with the reviewer. We have included a mention of this article in the introduction (2nd paragraph), which echoes our recommendations in future studies (discussion).
Line 57 : "commercially" - needs a "to the" commercially ....
Thanks for your comment, we have amended the manuscript.
Line 71 : "All participants gave their written consent prior to inclusion in the study, and their results were anonymised before analysis." - Where was it performed? Was there any IRB considering human study?
Thanks for your comment, we have amended the manuscript.
Line 73 : "2.1. Study procedures" - It's not clear if the AG is the game with the HMD or if the HMD is an additional step. Make it clearer: active gaming was performed with HMD (brand, year, country, weight: ....g)
Thanks for your comment, we have amended the manuscript for enhanced accuracy regarding equipment used and conditions.
Line 79 : "During this testing session, participants performed three tests in a randomized order: a submaximal walking test on a treadmill at 6 km·h-1 (W6)" - Please describe better how the tests were carried-out. How much time each, environment, temp (if controlled) etc...
Thanks for your comment, we have amended the manuscript for enhanced accuracy regarding procedures.
Equation n°1 : "%HR = 208-0.7×(age in year)" - Its not clear why is this formula for. "HR was expressed relative to individual age-predicted maximal HR (HR%) using equation (1) [18]: "This formula is used to base the training in a sub maximal HR, but is %HR how close they reached the maximal?
Thanks for your comment, we have amended the manuscript regarding this section.
Equation n°2 : " EE = VO2×(5×RER+16.1)×60" - Just better clarify this. VCO2/VO2 was measured? Does the "5x" in the formula, means it was performed 5 times? I'm not an expert and I think the formula should be explained in every step so it's clear the reason of this. Is the RER calculation an step non described? if so, please describe it.
Thanks for your comment, which echoes that of reviewer 1. We have improved this section and added a reference to this section.
The measure of RER in and of itself was not a variable of interest for our study. We used RER in the calculation of EE. As such, we chose not to report it as a result for clarity's sake.
The energy expenditure equation we used is a standard calculation in exercise physiology using the linear regression of EE as a function of RER and VO2 (with a y=ax+b form with known energy equivalents of lipids and glucids oxidized at different RER levels). We have added the reference of the princeps study on substrate utilization as a function of RER. We have updated the manuscript accordingly.
Line 112 : "Bonferroni post-hoc"
Thanks for your comment, we have amended the manuscript accordingly.
Line 130 : "W6" - In the table is walk / Table 1 : "Walk" - maybe let it W6, once the name of the component.
Thanks for your comment, we have amended the manuscript accordingly.
Table 1 : "HR% (%)" - % of %? Because HR% it's already in percentage, or not?
Thanks for your comment, we have amended the manuscript accordingly.
Line 145 : "differences" - on HR%? clarify.
Thanks for your comment, we have amended the manuscript accordingly.
Line 192 : "and" - "but", maybe? this sentence is unclear.
Thanks for your comment, we have amended the manuscript accordingly.
Line 197 : " Therefore, adding weights at the wrists induced EE comparable to walking only at higher AG levels" - It's rather confusing with "Therefore" so .. just a suggestion.. However, adding weights at the wrists only induced EE changes comparable to W6 at higher levels...
Thanks for your comment, we have amended the manuscript accordingly.
Line 218 : "Still" – Yet.
Thanks for your comment, and for all previous ones. We appreciate the time and energy spent reviewing our manuscript and feel it has greatly improved its quality.

Round 2
Reviewer 1 Report
Manuscript is improved.
Line 18-19 abstract is a concern. May provide effect size or similar as there was no difference so why saying effective means of increasing energy expenditure?
Abstract Line 21. What is meant by enhanced immersive feelings provided during. What data is this statement based upon?
Again the abstract conclusions do not match results statement. Abstract needs updated / revised.
Relative Energy Expenditure should be used throughout. EE was not directly measured.
Author Response
RESPONSES TO REVIEWERS
We thank the reviewer for their quick and thorough review. Please find attached our revised manuscript taking into account all the recommendations and suggestions.
Reviewer #1 (Comments to the Author (Required)):
Line 18-19 abstract is a concern. May provide effect size or similar as there was no difference so why saying effective means of increasing energy expenditure?
The reviewer is correct, thanks for pointing this out. Adding weights had an effect only when looking at the effect of game level (at maximum reached), but no effect on average. We have reworded this more carefully.
“We observed that playing AG with wrist-worn weights could be used as a means of increasing energy expenditure only at maximum game level, but ineffective otherwise.”
Abstract Line 21. What is meant by enhanced immersive feelings provided during. What data is this statement based upon?
Thanks for your remark, and we apologise for not catching this unfinished sentence. We have reworded it to remove any speculative aspect, and have used a more specific wording.
“Our findings indicate that AG in a HMD may not provide a sufficient stimulus to meet recommended physical activity levels despite increased psychophysiological load.”
Again the abstract conclusions do not match results statement. Abstract needs updated / revised.
Thanks for this, we agree and have amended the section.
“Surprisingly, adding a simple constraint (wrist-worn weights) during active gaming was effective to increase energy expenditure at levels comparable to conventional exercise only at the highest level of the game.”
Relative Energy Expenditure should be used throughout. EE was not directly measured.
Thanks for your remark, agreed. We have amended the manuscript using EE when possible as it was already defined, modified it to “relative energy expenditure” when relevant, and kept using “energy expenditure” for general remarks (as in the introduction/discussion/conclusion).
